*The Company of*
**Biologists**

## RESEARCH ARTICLE

# A functional comparison of two transplantable syngeneic mouse models of melanoma: B16F0 and YUMM1.7

David J. Klinke, II[1,*], Alanna Gould[2], Anika Pirkey[1], Atefeh Razazan[3] and Wentao Deng[4]

## ABSTRACT

The B16 murine melanoma cell lines are considered the gold standard for testing melanoma immunotherapies due to low treatment success rates. However, the clinical relevance of these models has been questioned due to a mutational landscape void of driver mutations typically seen in human melanomas and a tendency to form necrotic cores at high tumor volumes. Creating the YUMM1.7 line addressed these limitations by providing an additional contextually consistent model with a more clinically relevant genetic background. The combined use of both models can generate stronger studies in melanoma immunology and immunotherapy. However, to date, there have been no direct functional comparisons of the characteristics of these two models to inform the design of such studies. To address this, we conducted a series of functional experiments to characterize the kinetics of tumor growth, chemotherapeutic sensitivity, and immunogenicity of these models. We found that the B16F0 model had faster intrinsic tumor growth rates, was more susceptible to lysis by tumor-specific CD8+ T cells, and secreted higher levels of the angiogenic factors VEGF and Ang2. Meanwhile, the YUMM1.7 model was more sensitive to chemotherapeutic treatment, secreted higher levels of chemokines CCL2, CXCL1, and CX3CL1, and showed higher infiltration of lymphocyte and myeloid subsets at the same tumor size. Overall, YUMM1.7 model may be better suited for *in vivo* studies of mechanisms that require a wider observation window and intervention than the B16F0 model, such as immune response. However, angiogenesis and immunotherapy studies may benefit from a more in-depth comparative analyses of both models.

KEY WORDS: *In vivo* mouse tumor models, Cancer immunology, Markov chain Monte Carlo

## INTRODUCTION

Murine cancer cell lines are widely used to study the interaction between malignant cells and host immunity *in vivo* (Connolly et al., 2022) with the B16 cell lines being some of the most frequently used. These clonal cell lines were isolated by Dr. Isaiah J. Fidler in

the mid-1970s and are derived from a spontaneously arising melanoma in an aged male C57BL/6 mouse (Fidler, 1975). Different subclones were derived based on their relative propensity to metastasize, with B16F0 having the least potential and B16F10 having the most potential to metastasize. These B16 variant cell lines are used to study melanoma cell biology, metastasis, and immune evasion mechanisms. The B16 cell lines have long been thought of as the gold standard of immunocompetent murine melanoma models, as they provide a tough test of immunotherapy (Overwijk and Restifo, 2001; Ya et al., 2015). However, with the emergence of small molecule drugs for melanoma targeting BRAF mutations, which are present in over half of melanoma cases (Pollock et al., 2003), the clinical relevance of a cell line with wild-type BRAF was brought into question as the mutational landscape of the B16 line is unlike that seen in human melanoma (Zhong et al., 2020) particularly in the context of common driver mutations.

To address this limitation, the Yale University Mouse Melanoma (YUMM) cell lines were developed as transplantable cell lines with defined genetic alterations. In particular, the YUMM1.7 cell line is derived from a 4-hydroxytamoxifen-induced melanoma tumor in a male C57BL/6 mouse (Meeth et al., 2016). This cell line was genetically engineered to be homozygous negative for wild-type PTEN and CDKN2 and harbors the BRAF V600E mutation. YUMM1.7 cells can be efficiently used for basic science studies of melanoma biology and immunology and to identify and evaluate new immunotherapies. The YUMM1.7 model offers researchers additional diversity with a more human-relevant genetic background for cancer immunology studies. While both the YUMM1.7 and B16F0 cell lines provide pre-clinical options for testing novel therapies, together they provide a more circumspect evaluation. Yet to date, there have been no direct functional comparisons of the characteristics of these two models to inform the design of such studies. In this work, we will characterize the kinetics of tumor progression, chemotherapeutic sensitivity, and immune response of these two cell lines to provide context for how future works can best leverage their similarities and differences to generate robust results in pre-clinical immunology and immunotherapy studies.

## RESULTS AND DISCUSSION

### *In vivo* tumor growth kinetics in C57BL/6 and NSG mice

To isolate the intrinsic growth rate of each tumor model from the selective pressure of host immunity, we compared tumor growth of the same cell line in two mouse strains that differ in host immune response. NOD.Cg-Prkdc[*scid*] Il2rg[*tm1Wjl*]/SzJ (NSG) mice lack functional NK cells, T cells, and B cells, enabling tumors to grow at their maximum intrinsic rate without the selective pressure of host immunity. C57BL/6 mice have intact immune systems that can recognize and kill tumor cells, which can slow down tumor growth. One of the barriers commonly cited for engaging anti-tumor

[1]Department of Chemical and Biomedical Engineering, West Virginia University, Morgantown, MV 26506, USA. [2]Department of Biochemistry, The Medical College of Wisconsin, Milwaukee, WI 53226, USA. [3]Department of Physiology, Pharmacology and Toxicology, West Virginia University, Morgantown, WV 26506, USA. [4]Department of Genomic Medicine, The MD Anderson Cancer Center at University of Texas, Houston, TX 77030, USA.

*Author for correspondence (david.klinke@mail.wvu.edu)

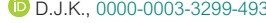 D.J.K., 0000-0003-3299-4938

immunity is that tumor cells lack antigens that the immune system recognize as foreign. In theory, sex mismatch in transplanting male cells, both lines originated in male mice, containing HY antigens in female C57BL/6 mice introduces a foreign antigen and enhances the immunogenicity of the tumor. By comparing tumor growth rates in these two hosts, we can calculate how much the immune systems affects each tumor model. To this end, immunocompetent C57BL/6 female mice and severely immunocompromised NSG mice were subcutaneously injected with wild-type B16F0 or YUMM1.7 tumor cells. B16F0 tumors grew rapidly, reaching 1000 mm$^3$ in size within 3 weeks in C57BL/6 and 2 weeks in NSG (Fig. 1A). In contrast, YUMM1.7 treated animals succumbed to a large progressive tumor in just over 4 weeks (Fig. 1B). Both cell lines showed log-linear growth in both C57BL/6 and NSG mice. We mathematically modeled tumor size as a function of the initial bolus size of tumor-initiating cells ($C_{T0}$) and the net growth rate ($k=k_P-k_D$) (Fig. 1C). Tumor growth in severely immunocompromised NSG mice was used to estimate intrinsic proliferation rate constants ($k_P$) for both cell lines (Fig. 1D). We noted that the $k_P$ of the B16F0 line was significantly higher than the YUMM1.7 line (median $k_P$ YUMM1.7=0.44 day$^{-1}$ versus B16F0=0.65 day$^{-1}$). In the immunocompetent C57BL/6 mice, the net proliferation rate reflects both intrinsic growth and immune-related cell death ($k_D$). In this mathematical model, one parameter, $k_D$, captures all anti-tumor immunity mechanisms present in

C57BL/6 mice but absent in NSG mice. More importantly, the value of this one parameter can be uniquely determined by the available data, as indicated by the bounded posterior distributions (Fig. 1E). The difference in immune-related cell death can then be inferred by comparing the net growth rate of tumors initiated by injecting aliquots of a cell line into NSG versus C57BL/6 mice, which is expressed as a log-ratio (Fig. 1E). A log-ratio equal to zero implies that the net growth rate of tumors in NSG versus C57BL/6 is the same while a log-ratio value greater than zero implies that tumors derived from the cell line grow faster in severely immunocompromised NSG than C57BL/6 mice. The YUMM1.7 cell line exhibited lower log-ratios than the B16F0 cell line (median log-ratio B16F0=0.195 versus median log-ratio YUMM1.7=−0.009). In addition, net tumor growth rates of the YUMM1.7 cell line in C57BL/6 and NSG mice were not statistically different, as evident by the posterior distribution of the log-ratio being centered around 0, while the growth rate of the B16F0 model was significantly higher in NSG mice. Both of these findings are consistent with literature citing that genetically engineered models are typically less immunogenic than spontaneous tumor models induced by carcinogens (DuPage and Jacks, 2013; Lee et al., 2016; McFadden et al., 2016; Connolly et al., 2022). Despite the presence of HY antigen, the reduced immunogenicity of the YUMM1.7 cell line observed by Bosenberg and colleagues motivated exposure to UV radiation creating the YUMMER1.7 derivative (Wang et al., 2017).

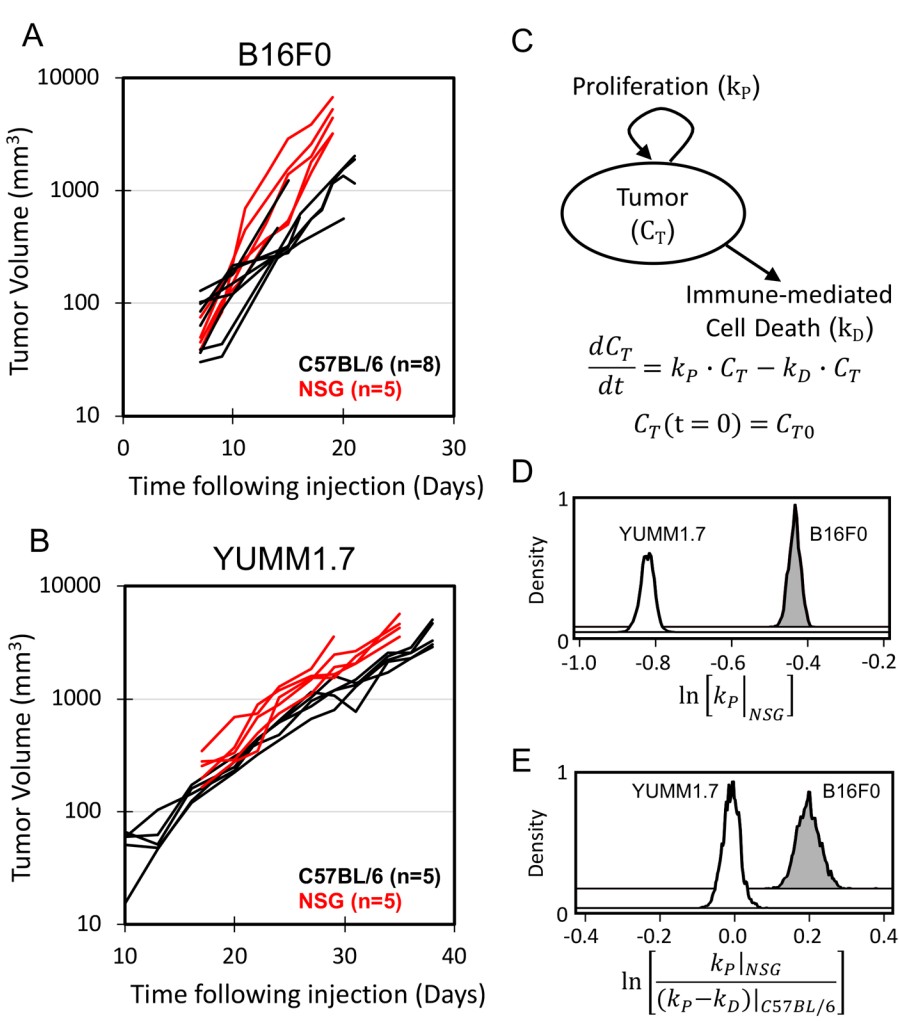

Fig. 1. B16F0 tumors have a faster intrinsic growth rate and are more immunogenic than YUMM1.7 tumors. Growth curves of B16F0 (A) and YUMM1.7 (B) tumors in immunocompetent C57BL/6 (black) and severely immunocompromised NSG (red) mice. Each curve corresponds to the tumor size in each mouse measured as a function of time. A mathematical model describing the growth kinetics (C) was regressed to the log-linear growth curves using a Markov chain Monte Carlo approach. (D) The posterior distribution in the parameter corresponding to the proliferation rate ($k_P$) of YUMM1.7 (unshaded) and B16F0 (gray shaded) cell lines in NSG mice. (E) The posterior distribution in the log ratio of the proliferation rate in NSG mice ($k_P$) relative to the net proliferation rate ($k_P-k_D$) of the same cells in C57BL/6 mice. The posterior distribution for YUMM1.7 cells is shown as an unshaded curve and for B16F0 cells in shown as a shaded curve.

The faster intrinsic growth rate of the B16F0 over the YUMM1.7 tumors *in vivo* has practical implications for experimental design. Therapies need time to elicit anti-tumor responses before changes in tumor size become measurable. For instance, when a therapy is designed to elicit a new adaptive anti-tumor immune response, it takes up to 14 days to fully activate the adaptive immune system (Miao et al., 2010). By this time, B16F0 tumors exceed 1000 mm$^3$ and develop ulcers and necrotic cores (Ya et al., 2015). These issues persist despite our use of female C57BL/6 mice, which has been shown to decrease B16F0 tumor volume and increase immune cell infiltration in comparison to male mice (Dakup et al., 2020). These complications shorten the observation window. On average, B16F0 and YUMM1.7 tumor volumes in C57BL/6 mice reached approximately 1000 mm$^3$ by days 18 and 28, respectively, with all mice in the B16F0 cohorts typically requiring humane euthanasia by 21 days. In contrast, the majority of mice in the YUMM1.7 cohorts lasted up to 38 days. Even at the end points, ulceration and necrotic cores were not prevalent in mice bearing YUMM1.7 tumors.

In addition to this short observation window for mechanistic study, most experimental immunotherapies published are only able to show successful slowing or delaying of tumor growth in B16F0 tumors when treatment is administered prior to day 5 post-inoculation, which is typically before palpable tumors present (Wen et al., 2012). Despite slower tumor establishment, the YUMM1.7 model provides a better experimental timeline for studying mechanisms with slower response times, like engaging a primary immune response.

### Impact of chemotherapeutic drugs on cell viability

Given that chemotherapy drugs have been used as a first-line treatment for melanoma, we first tested the impact of three chemotherapy drugs on the viability of B16F0 and YUMM1.7 cell lines. The three drug panel included PLX-4720, an inhibitor of BRAFV600E; iCRT14, a Wnt pathway inhibitor; and mitoxantrone, a chemotherapy drug that causes DNA damage, apoptosis, and immunologic cell death (Fujimoto and Ogawa, 1982; Sukkurwala et al., 2014). These three drugs were chosen as they span from targeting specific molecular pathways (PLX-4720 and iCRT14) to exhibiting a broad spectrum of activity (mitoxantrone). B16F0 and YUMM1.7 cells were exposed to PLX-4720 at concentrations ranging from 0-12 µM/ml, mitoxantrone from 0-200 nM/ml, or iCRT14 from 0-25 µM/ml for 24 h. Our results showed that there was a prominent dose-dependent decrease in cell viability in both treated cell lines after 24 h. Among the three small molecule drugs, PLX-4720 and mitoxantrone had a differential effect between the two cell lines. The IC50's for PLX-4720 were almost a factor of 100 different, with 138.6 µM for B16F0 and 1.734 µM for YUMM1.7. This was expected, as the YUMM1.7 cell line was derived from mice harboring the activating BRAFV600E mutation inhibited by PLX-4720 while the B16F0 line retains wild-type BRAF. Mitoxantrone had a slightly less differential effect than PLX-4720, with observed IC50s of 68.07 nM for B16F0 and 9.528 nM for YUMM1.7 cells. While iCRT14 also had a significant dose-dependent effect on viability, the IC50s were similar with values of 3.43 µM for B16F0 and 3.37 µM for YUMM1.7 cell lines (Fig. 2). In terms of relative toxicity, mitoxantrone was the most toxic with IC50's in the nanomolar range compared to the micromolar range for iCRT14 and PLX-4720. It is not surprising that mitoxantrone was the most toxic, given its broad spectrum of activity. It is interesting to note that iCRT14 exhibited a similar level of toxicity in both cell lines as PLX-4720 had on cells harboring the

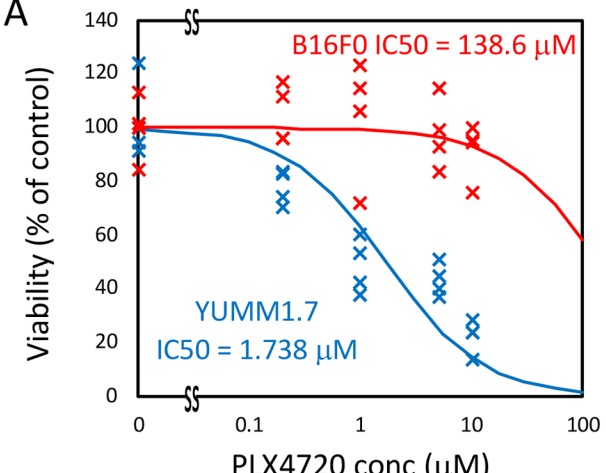

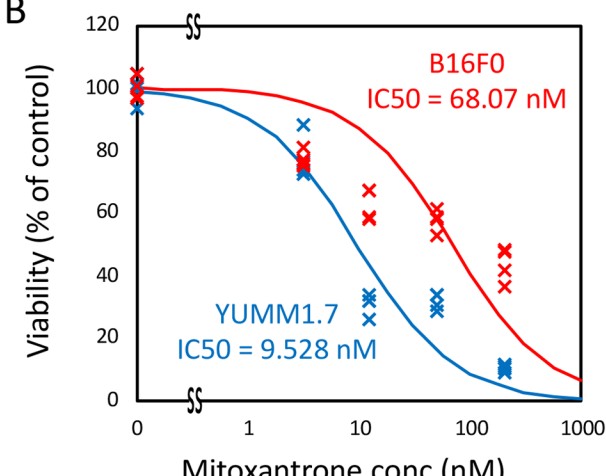

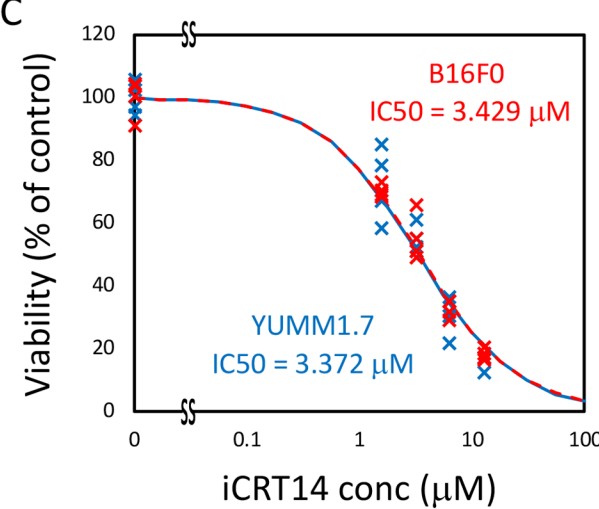

**Fig. 2. Viability of B16F0 and YUMM1.7 cells in response to increasing concentrations of three small molecule drugs.** Viability of YUMM1.7 (blue) and B16F0 (red) cells cultured *in vitro* with the indicated concentrations of PLX4720 (A), Mitoxantrone (B), and iCRT14 (C). Symbols represent results from four biological replicates at each concentration. The IC50s indicated were calculated by regressing dose-response curves (100×(1−$x$/($x$+IC50)), where $x$ is the drug concentration), which are shown as curved lines, to data for each drug-cell combination.

activating BRAFV600E mutation, which was engineered into the YUMM1.7 line.

## Sensitivity to antigen-specific CD8+ T cell responses

CD8+ T lymphocytes were isolated and purified from PMEL1 C57BL/6 mice and pre-activated with PMEL1 CD8+ T cell specific peptide (hgp100$_{25-33}$) for 2 days. These pre-activated PMEL1 CD8+ T cells were then co-cultured with target cells at the indicated effector-to-target ratio for 4 h. Using a cytotoxicity assay, both targeted cell lines exhibited an increase in cytotoxicity with increasing effector-to-target ratio. Lysis of B16F0 cells by hgp100$_{25-33}$-activated T lymphocytes was significantly higher than that of YUMM1.7 cells across all effector-to-target ratios (Fig. 3, P-value≤0.01). Next, we explored possible reasons for this difference.

The pro-inflammatory cytokine IFN-$\gamma$ plays an important role in regulating immune responses within the tumor microenvironment (TME). IFN-$\gamma$ directly acts on tumor cells to induce MHC class I expression. Studies show that exposure to rmIFN-$\gamma$ *in vitro* increases surface expression of MHC class I on tumor cells, thus potentially increasing their recognition by cytotoxic CD8+ T lymphocytes. To compare the extent of this effect in the two cell lines, wild-type B16F0 and YUMM1.7 cell lines were incubated with or without 250 U/ml of rmIFN-$\gamma$. Under normal culture conditions, unstimulated B16F0 and YUMM1.7 cells expressed statistically similar low levels of H2K$^b$ and H2D$^b$ (Fig. 4 left column, rows B and C). The B16F0 cell line expressed significantly higher levels of PD-L1 at baseline than the YUMM1.7 line, but levels were still relatively low (Fig. 4 left column, row A). Both cell lines upregulated PD-L1, H2K$^b$ and H2D$^b$ after IFN-$\gamma$ stimulation, though the difference in expression levels was greater in B16F0 than in YUMM1.7 (Fig. 4 middle column).

## Relative abundance of tumor-infiltrating immune subsets and tumor-secreted factors

To characterize the differences in immune cell infiltration to the different cell lines, single cell suspensions from tumors generated

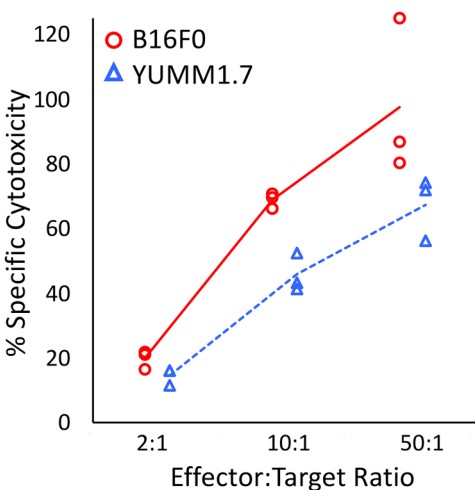

**Fig. 3. Primary PMEL1 CD8+ T cells lyse both B16F0 and YUMM1.7 cells.** Pre-activated PMEL1 CD8+ T cells were co-cultured for 4 h with target cells at the indicated effector to target ratios to assess cytotoxicity. Cytotoxicity was quantified using a CyQUANT LDH Cytotoxicity assay. Wild-type B16F0 and YUMM1.7 cell lines were used as targets after an overnight incubation with 250 U/ml of rmIFN$\gamma$. Collective results from three biological replicates. Statistical significance of a difference between the two cell lines was assessed using a two-factor ANOVA with replication.

by inoculating C57BL/6 mice with wild-type B16F0 or YUMM1.7 cells were aliquoted among three antibody panels and characterized via flow cytometry as described previously (Fernandez et al., 2022). Among tumors of similar sizes, B16F0 tumors showed significantly lower CD45+ cell infiltration compared to YUMM1.7 tumors (Fig. 5A, P-value≤0.01). The relative prevalence of NK cells, CD8+ T-cells, and CD4+ T-cells was higher in the YUMM1.7 cell line than the B16F0 line (P-value for NK cells≤0.01, CD8+ T-cells≤0.05, CD4+ T-cells≤0.01). In contrast, B-cells were significantly more prevalent in the B16F0 tumors compared to the YUMM1.7 tumors (Fig. 5B, P-value≤0.001). In addition to assaying changes in T, B, and NK cells, we also reviewed changes in the myeloid compartment, focusing on CD11c+ and CD11c− macrophages and three different myeloid derived suppressor cell (MDSC) populations: CD11c+ monocytic (CD11c+ Mo-MDSC), CD11c− monocytic (Mo-MDSC), and polymorphonuclear (PMN-MDSC) (Fig. 5C). Mo-MDSCs and CD11c− macrophages were significantly less prevalent in YUMM1.7 tumors compared to B16F0 tumors (P-values≤0.01) while CD11c+ macrophages, CD11c+ Mo-MDSC, and PMN-MDSCs were significantly more prevalent in the YUMM1.7 tumors (P-values CD11+ macs and CD11C+ Mo-MDSCs≤0.01, PMN-MDSCs≤0.05).

To identify potential secreted factors that underpin the observed differences in immune cell infiltration between these two tumor models, we measured cytokine, chemokine, and growth factor production by wild-type YUMM1.7 and B16F0 cells *in vitro* using the R&D Systems Mouse XL Cytokine Array (Fig. 6). The dotted lines enclose a null distribution determined by the values and uncertainty in quantifying the positive and negative controls on arrays exposed to media conditioned by YUMM1.7 versus B16F0 cells, that is differences in abundance explained by random chance. The vertical distance of a measured secreted factor from the x-axis relative to the vertical distance between the null distribution curve and the x-axis corresponds to a z-score. That is, as the vertical distance increases relative to the null distribution curve, the observed difference is increasingly unlikely to be explained by random chance alone. As differences in angiogenesis between B16F0 and YUMM1.7 tumors may underpin differences in total CD45+ cell infiltration, we found that conditioned media from B16F0 cells contained higher levels of the angiogenic factors vascular endothelial growth factor (VEGF) and angiopoietin-2 (Ang2) compared to that from YUMM1.7 cells. VEGF and Ang2 are known to promote the growth of new, immature blood vessels. In addition, Ang2 has been shown to inhibit blood vessel maturation and normalization by inhibiting Ang1 signaling via Tie2 in vascular endothelial cells (Yuan et al., 2009; Saharinen et al., 2011).

Interestingly, the angiogenic response to VEGF exhibits a biphasic response where angiogenesis increases with VEGF at low expression levels while high VEGF expression has an inhibitory effect (Carpentier et al., 2020). It is known that tumors formed by subcutaneous injection of B16 cells become necrotic when allowed to grow larger than 1000 mm³, which occurs within 14-21 days (Ya et al., 2015). Indeed, in our hands, YUMM models tend to produce solid tumors with more developed blood vessels, while tumors derived from B16 models tend to be more avascular and necrotic. Taken together, the two models may provide an interesting context to explore how modulating these different angiogenic factors, VEGF and Ang2, influence immune cell infiltration within tumors, a first step in engaging an anti-tumor immune response.

Within the CD45+ compartment, observed differences among immune cell composition may be explained by chemokine and

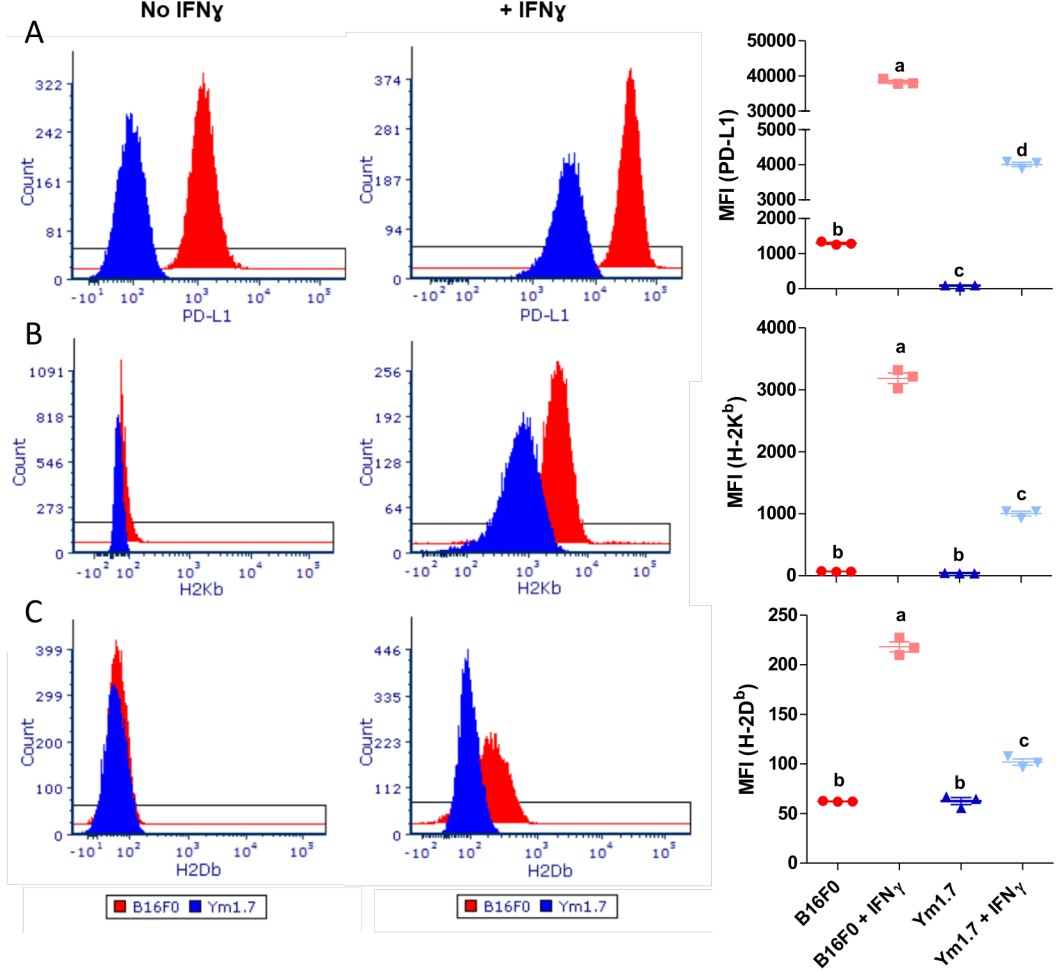

**Fig. 4. IFN-gamma induces PD-L1 and MHC class I expression in both B16F0 and YUMM1.7 cells.** Expression levels of PD-L1 (A, top row), H-2K$^b$ (B, middle row), and H-2D$^b$ (C, bottom row) in B16F0 (red) and YUMM1.7 (blue) were quantified by flow cytometry at baseline (left column) and after overnight incubation with 250 U/ml of IFN$\gamma$ (middle column). Dot plots (right column) summarize differences in mean values of distributions among cell lines and conditions. Mean±s.d. represent collective results from three biological replicates. ANOVA with post-hoc Tukey tests were used to assess statistical significance, where (a, b, c, or d) denote statistically different groups.

cytokine expression differences. Notably, the chemokines CCL2, CXCL1, and CX3CL1 were upregulated in the conditioned media from YUMM1.7 cells compared to that from B16F0 cells. CCL2 and CXCL1 have previously been associated with MDSC infiltration into tumors via their respective receptors, CCR2 and CXCR2 (Huang et al., 2007; Taki et al., 2018). CX3CL1 has been

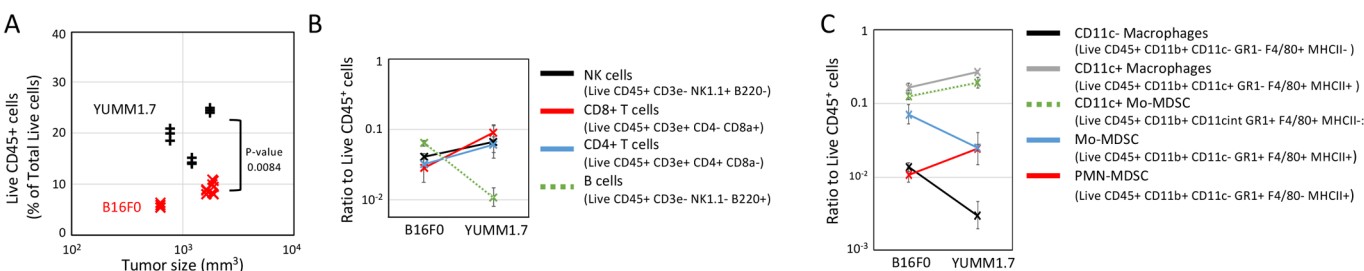

**Fig. 5. Tumors derived from YUMM1.7 cells have higher infiltration of lymphocytes compared to B16F0-derived tumors.** (A) The percentage of live CD45+ cells isolated from tumors generated by inoculating subcutaneously, with WT B16F0 (red) and YUMM1.7 (black) cells. CD45+ values were obtained from three different antibody panels that quantified T cells, B/NK cells, and myeloid cells in TIL isolates (considered a technical replicate) from each mouse (considered a biological replicate) (B16F0 *n*=3 technical replicates of four biological replicates; YUMM1.7 *n*=3 technical replicates of three biological replicates). *P*-values calculated using two-sided Student's *t*-test with equal variance and biological replicates averaged across technical replicates. (B) A comparison of the ratio of NK cells (black), CD8+ T cells (red), CD4+ T cells (blue), and B cells (dotted green) to live CD45+ TILs in tumors generated using WT B16F0 and YUMM1.7 cells (mean±s.d.). (C) A comparison of the ratio of CD11c– (black) and CD11c+ (gray) macrophages, CD11c+ Mo-MDSCs (dotted green), Mo-MDSCs (blue), and PMN-MDSCs (red) to live CD45+ TILs in tumors generated using WT B16F0 and YUMM1.7 cells (mean±s.d.). A statistically significant difference in ratio between YUMM1.7 and B16F0 TIL subset was assessed using a two-tailed, unpaired Student's *t*-test with equal variance (see text for values).

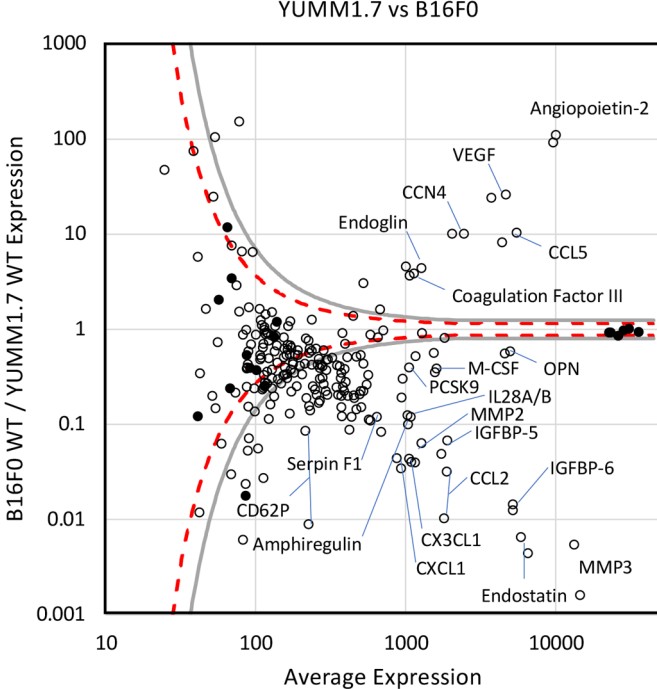

**Fig. 6. The profile of cytokines, chemokines, and growth factors secreted into media conditioned by WT B16F0 and YUMM1.7 are different.** Results from R&D Systems' Mouse XL Cytokine Array Kit of cytokine, chemokine, and growth factor expression by WT YUMM1.7 and B16F0 cells *in vitro*. In presenting the combined results for assaying each cell line's secretome using a single biological replicate, open circles represent results for specific cytokine probes, which are spotted in duplicate on the array, and filled circles represent positive and negative controls. Dotted lines enclose 95% of a null distribution estimated from positive and negative controls. Gray shaded lines indicate Z-scores of 3 and −3. Particular secreted factors that are differentially expressed are annotated with their respective names.

associated with the accumulation of tumor-associated macrophages (TAMs) and their polarization toward a protumor M2-like phenotype (Ishida et al., 2020; Schmall et al., 2015). Meanwhile, conditioned media from B16F0 cells contained higher levels of CCL5, which has been associated with the recruitment and activation of TAMs, MDSCs and T regulatory cells (T-regs) (Aldinucci et al., 2020; Walens et al., 2019; Bai et al., 2020; Melese et al., 2022; Chang et al., 2012). While these immune cell subsets do not all map onto the cell subsets profiled by our flow cytometry panel (Fig. 5), we noted that the only myeloid subsets that showed significantly higher infiltration in B16F0 tumors than YUMM1.7 tumors were CD11c− macrophages and Mo-MDSCs, while YUMM1.7 tumors contained more CD11c+ macrophages and MDSCs. This may be explained by the increased abundance of CCL2 in YUMM1.7 tumors. CCL2 has been shown to promote adhesion of monocytes to vascular endothelial cells by increasing surface expression of CD11b and CD11c on monocytes. It has also been shown to promote the differentiation of monocytes into MDSCs (Gschwandtner et al., 2019). Additionally, macrophage colony stimulating factor (M-CSF/CSF-1), which is also slightly upregulated in YUMM1.7 cells, has been shown to bind the CSF-1 receptor expressed by CD11c+ macrophages and, in combination with CCL2, promote their polarization toward an immunosuppressive M2-like phenotype (Sierra-Filardi et al., 2014).

## Conclusions

Collectively, our functional comparison of these two transplantable models for melanoma is summarized in Table 1. The B16F0 cell line has long been used as the benchmark for pre-clinical testing of melanoma immunotherapies due to the difficult bar it sets for success (Overwijk and Restifo, 2001; Ya et al., 2015). However, the YUMM cell lines developed at Yale University offer an additional possibility that addresses the lack of similarity in the mutational landscape between human and murine melanoma while holding some benefits over the B16F0 line in certain contexts (Zhong et al., 2020; Meeth et al., 2016). In comparing the two cell lines directly, the B16F0 cell line has a faster intrinsic growth rate than the YUMM1.7 line. When combined with its tendency to form a necrotic core and ulcerate at high tumor volumes (Ya et al., 2015), this fast growth rate has practical implications as it can lead to difficulty in studying mechanisms that require longer response times, such as eliciting a primary adaptive immune response. In contrast, the YUMM1.7 model takes longer to form palpable tumors, but offers a wider window of opportunity to study these slower mechanisms because the mouse model's health is not jeopardized by necrosis or ulcerations that lead to early euthanasia.

The lack of necrotic core in the YUMM1.7 model may be due to changes in the secreted factors present between the two models. The B16F0 model shows higher levels of VEGF and Ang2 in conditioned media, potentially leading to the formation of immature blood vessels or the inhibition of vessel formation (Yuan et al., 2009; Saharinen et al., 2011; Carpentier et al., 2020). Studies of angiogenesis in melanoma tumors could benefit from a comparative analysis of these two cell lines. In particular, modulating secreted factors to improve angiogenesis may provide a path to enhance the efficacy of cancer immunotherapies by increasing CD45+ immune infiltration, which was lower in the B16F0 tumors compared with YUMM1.7 tumors. Clinically, the presence of immune cell infiltration is a strong independent predictor of a positive response to immune checkpoint therapy in melanoma (Tumeh et al., 2014; Kümpers et al., 2019). The reduced tumor infiltrating lymphocytes (TILs) in the B16F0 model suggests that it is not likely to respond to immune checkpoint therapies, as has been reported (Hackett et al., 2022; Singh et al., 2014). Understanding the intersection of angiogenesis and immune infiltration may provide insight into potential mechanisms at work in human melanoma.

The increased immune infiltration, including NK and CD8+ T cells, in YUMM1.7 tumors suggests a more immunologically "hot" tumor microenvironment; yet, this was unexpected. As comparing the growth of tumors in NSG to C57BL/6 mice, it appeared that the growth of YUMM1.7 tumors was not impacted by host immunity. The presence of immune cells that do not appear to kill tumor cells suggests local factors that blunt immune-mediated cytotoxicity, such as tumor-secreted CCN4 (Fernandez et al., 2022). While additional work may be required to further parse these complicated profiles of secreted proteins, these results highlight a more complicated and nuanced biology associated with two seemingly similar pre-clinical mouse models for melanoma. Moreover, clarifying the mechanisms that underpin the phenotypes observed in these mouse models may provide insight into the heterogeneity of human melanoma and a path towards new treatment options.

## MATERIALS AND METHODS
### Mice
6- to 8-week-old female C57BL/6Ncrl and NOD.Cg-Prkdc*scid* Il2rg*tm1Wjl*/SzJ (NSG) mice were purchased from Charles River Laboratories and The

**Table 1. Summary of functional comparison of B16F0 and YUMM1.7 melanoma models**

| Characteristic | B16F0 | YUMM1.7 |
|---|---|---|
| Genotype/phenotype | | |
| Genetic background | Wild-type BRAF | $BRAF^{V\,600E}$, $PTEN^{-/-}$, $CDKN2^{-/-}$ |
| Clinical relevance | Limited (lacks driver mutations) | **Higher** (human-relevant mutations) |
| Intrinsic growth rate ($k_P$) | **Faster** ($0.65$ day$^{-1}$) | **Slower** ($0.44$ day$^{-1}$) |
| Time to 1000 mm$^3$ | ~18 days (C57BL/6) | ~28 days (C57BL/6) |
| Maximum experimental window | Limited (~21 days) | Extended (~38 days) |
| Necrotic core formation | **Frequent** at >1000 mm$^3$ | **Rare** even at endpoint |
| Ulceration | Common by day 21 | Uncommon even at 38 days |
| Vascularization quality | Avascular, necrotic | Well-developed vessels |
| Immunogenicity | | |
| Immune selective pressure (log-ratio k's) | **Significant** (0.195) | **Minimal** (−0.009) |
| Total CD45$^+$ infiltration | **Lower** | **Higher** ($P \leq 0.01$) |
| NK cell infiltration | Lower | **Higher** ($P \leq 0.01$) |
| CD8$^+$ T cell infiltration | Lower | **Higher** ($P \leq 0.05$) |
| CD4$^+$ T cell infiltration | Lower | **Higher** ($P \leq 0.01$) |
| B cell infiltration | **Higher** ($P \leq 0.001$) | Lower |
| CD11c$^-$ macrophages | **Higher** ($P \leq 0.01$) | Lower |
| CD11c$^+$ macrophages | Lower | **Higher** ($P \leq 0.01$) |
| Mo-MDSCs | **Higher** ($P \leq 0.01$) | Lower |
| PMN-MDSCs | Lower | **Higher** ($P \leq 0.05$) |
| T cell sensitivity | | |
| CD8$^+$ T cell cytotoxic sensitivity | **Higher** ($P \leq 0.01$) | Lower |
| PMEL1T cell recognition | Yes, enhanced | Yes, but reduced |
| Antigen presentation | | |
| Baseline MHC-I expression | Low | Low |
| IFN-$\gamma$ MHC-I upregulation | **Higher** response | Moderate response |
| Baseline PD-L1 expression | **Higher** | Lower |
| Secreted angiogenic factors | | |
| VEGF | **Higher** | Lower |
| Angiopoietin-2 | **Higher** | Lower |
| Secreted chemokines/cytokines | | |
| CCL2 | Lower | **Higher** |
| CCL5 | **Higher** | Lower |
| CXCL1 | Lower | **Higher** |
| CX3CL1 | Lower | **Higher** |
| Drug sensitivity | | |
| PLX-4720 sensitivity (IC$_{50}$) | 138.6 µM (resistant) | 1.738 µM (**sensitive**) |
| Mitoxantrone sensitivity (IC$_{50}$) | 68.07 nM | 9.528 nM (**more sensitive**) |
| iCRT14 sensitivity (IC$_{50}$) | 3.429 µM | 3.372 µM (similar) |

Jackson Laboratory, respectively, and randomly assigned to treatment groups for tumor growth assays. 8- to 12-week-old female transgenic B6.Cg-Thy1a/Cy Tg(TcraTcrb)8Rest/J (PMEL1) mice were purchased from The Jackson Laboratory and used as a source of transgenic CD8+ T cells. All animal experiments were approved by the West Virginia University (WVU) Institutional Animal Care and Use Committee and performed on-site (IACUC Protocol #1604002138). Mice were co-housed in sterilized micro-isolator cages and facility sentinel animals were regularly screened for specific pathogenic agents. No animals were excluded from reporting of the study. As is custom in the field, investigators were not blinded to cohort membership when measuring tumor size.

### Reagents and cell culture

The mouse melanoma line B16F0 (purchased in 2008, RRID: CVCL_0604) was obtained from American Tissue Culture Collection (ATCC, Manassas, VA, USA). The mouse melanoma line YUMM1.7 (received in September 2017, RRID: CVCL_JK16) was a gift from Drs. William E. Damsky and Marcus W. Bosenberg (Yale University) (Meeth et al., 2016). The cells were cultured at 37°C in 5% CO$_2$ in high-glucose DMEM (Cellgro/Corning, NY, USA) supplemented with L-glutamine (Lonza, NJ, USA), 1% penicillin-streptomycin (Gibco, Thermo Fisher Scientific, MA, USA), and 10% heat-inactivated fetal bovine serum (Hyclone). All cell lines were revived from frozen stock, used within 10-15 passages that did not exceed a period of 6 months, and routinely tested for mycoplasma contamination by PCR.

Mitoxantrone hydrochloride, PLX-4720 and iCRT14 were purchased from Selleckchem, Cayman Chemical Co, and Sigma-Aldrich, respectively. The peptide hgp100$_{25-33}$ (KVPRNQDWL) was purchased from GenScript. Recombinant mouse IL-2 was purchased from eBioscience (Thermo Fisher Scientific, MA, USA). Recombinant mouse IFN-$\gamma$, PE-conjugated anti-mouse CD274 (PD-L1, clone 10F.9G2, dilution 1:20), PerCP/Cy5.5-conjugated anti-mouse H-2Db (clone KH95, dilution 1:20), and APC-conjugated anti-mouse H-2Kb (clone AF6-88.5, dilution 1:80) were purchased from BioLegend (San Diego, CA, USA).

### *In vivo* tumor growth

For tumor challenge, mice were implanted subcutaneously on the right flank with $3 \times 10^5$ B16F0 or $5 \times 10^5$ YUMM1.7 cells in PBS. Tumor growth was monitored every two days by measuring two perpendicular diameters of the tumor using calipers. The tumor volume was calculated according to the formulation ($\pi/6 \times$length$\times$width$^2$). For ethical considerations, mice were euthanized when tumor burden was significant enough to affect daily life (difficulty moving, weight loss, inability to feed or gain access to water, etc.).

### Flow cytometry

Four B16F0 and three YUMM1.7 tumors were surgically removed from C57BL/6 mice after euthanasia once tumors reached between 1000 and 1500 mm$^3$. Single-cell suspensions were obtained by enzymatically

digesting the excised tumors using the Tumor Dissociation Kit and gentleMACS C system (Miltenyi Biotec, Auburn, CA, USA). In addition to following the manufacturer's instructions, the gentleMACS program 37C_m_TDK_2 was used for YUMM1.7 tumors and 37C_m_TDK_1 was used for B16F0 tumors. Following red blood cell lysis, the remaining single-cell suspensions were washed and stained with Live/Dead Fixable Pacific Blue Dead Cell Stain Kit (Thermo Fisher Scientific) and blocked with Mouse BD Fc Block (BD Biosciences). Cell surfaces were then stained with one of three antibody mixes that focused on T cells (CD45, CD3, CD4, CD8, and PD1), NK and B cells (CD45, CD3, B220, NK11, DX5, and PD1), and myeloid cells (CD45, CD11b, CD11c, Gr-1, F4/80, and MHCII) and quantified by flow cytometry, as described in Fernandez et al. (2022). Specific details regarding the antibodies used are listed in Table S1.

### *In vitro* studies

For cell viability assays, B16F0 and YUMM1.7 were treated with Mitoxantrone, iCRT14, and PLX-4720 at the indicated concentrations for 24 h, 37°C and in 5% CO2. The cells were harvested and seeded in 96-well black plates with 5000 cells per well. Cell viability was determined using the ATPlite Luminesence ATP Detection Assay System (Perklin Elmer, USA) according to the manufacturer's instructions.

To assay antigen-specific cytotoxicity, splenocytes from PMEL1 mice were pre-activated *in vitro* for 2 days with 1 μM of the PMEL1 CD8+ T cell specific peptide [hgp100$_{25-33}$ (KVPRNQDWL)] in the presence of 100 U/ml of rmIL-2. On days two and four the cells were washed and re-cultured with fresh media containing 100 U/ml of rmIL-2. Pre-activated PMEL1 CD8+ T cells were co-cultured for 4 h with target cells, at the indicated effector to target ratios, to assess cytotoxicity that was quantified using a CyQUANT LDH Cytotoxicity assay (Invitrogen).

To assay PD-L1 and MHC class I expression, wild-type B16F0 and YUMM1.7 cell lines were stimulated directly *in vitro* in the presence and absence of rmIFN$\gamma$ (250 U/ml) for 24 h. Cells were washed twice with 200 μl fluorescence-activated cell sorting (FACS) wash buffer, pelleted by centrifugation at 2000 RPM for 5 min and resuspended in 200 μl of FACS wash buffer before monoclonal antibody staining. Cells were stained in the dark at 4°C for 30 min by using a cocktail of fluorochrome-conjugated anti-mouse antibodies for PD-L1/PE, H-2K$^b$/APC and H-2D$^b$/PerCP-Cy5.5. Stained cells were washed in flow wash buffer and fixed in 1% paraformaldehyde for 1 h at 4°C. The events were acquired using a BD LSR Fortessa (BD Biosciences) flow cytometry with FACSDiva software, where the fluorescence intensity for each parameter was reported as a pulse area with 18-bit resolution. Flow cytometric data were exported as FCS3.0 files and analyzed with FCS Express 6.0 (DeNovo Software, CA, USA) and R.

To collect tumor-conditioned media (TCM), wild-type B16F0 and YUMM1.7 cells were grown in complete DMEM until 80% confluency, washed with PBS (Cellgro/Corning, NY, USA) and incubated for 48 h in FBS-free DMEM. TCM were then centrifuged at 3000 *g* and 4°C for 15 min, and the supernatant collected and filtered. Cytokines, chemokines, and growth factors in TCM were detected with the Proteome Profiler Mouse XL Cytokine Array (R&D Systems, MN, USA), following the manufacturer's instructions.

### Statistical analysis

Results from all univariate experiments were analyzed by a two-tailed, unpaired Student's *t*-test. ANOVA was used to analyze multivariate experiments. A *P*-value of less than 0.05 was considered statistically significant. Tumor growth data were analyzed by estimating the growth rate of the tumors using a log-linear tumor growth model and a Markov chain Monte Carlo approach to generate the posterior distribution in the rate parameters, as described in Pirkey et al. (2023). In brief, we analyzed the tumor growth trajectory, which comprised between 3 and 8 B16F0 and 6 and 12 YUMM1.7 tumor size measurements obtained on different days, for each mouse separately but estimated a rate parameter collectively for a cohort. Following a burn-in period, parameter samples obtained from a converged Markov chain were used to estimate the posterior distribution. We used the log-ratio of growth rates in NSG versus C57BL/6 mice as a quantitative measure of immune pressure. A distribution in values greater than zero

indicate strong immune recognition, while a distribution in values that contain zero suggest immune evasion or tolerance. For cell viability assays, the IC50 s indicated were calculated for each drug-cell combination by regressing dose-response curves ($100\times(1-x/(x+IC50))$), where $x$ is the drug concentration). In analyzing the Cytokine Array, positive and negative controls were used to establish a null distribution, that is that an observed difference in abundance is explained by random chance. The curves enclosing the 95th percentile of the null distribution were calculated by regressing a curve to the twice the standard deviation among a subset of samples of a similar abundance versus the average of the sample subset. Statistical significance associated with differential expression can be estimated by comparing the vertical distance of a measured secreted factor from the x-axis relative to the vertical distance between the null distribution curve and the x-axis. The comparison of these two vertical distances is proportional to a z-score. Z-scores greater than 3 and less than −3 are considered significant and merit further consideration.

### Acknowledgements

The authors would like to thank Audry Fernandez for her assistance in conducting this study. Authors thank assistance by WVU Flow Cytometry & Single Cell Core Facility, RRID:SCR_017738.

### Competing interests

The authors declare no competing or financial interests.

### Author contributions

Conceptualization: D.J.K.; Data curation: D.J.K., A.G., A.R., W.D.; Formal analysis: D.J.K., A.P.; Funding acquisition: D.J.K.; Investigation: D.J.K., A.G.; Methodology: D.J.K.; Project administration: D.J.K.; Resources: D.J.K.; Software: D.J.K., A.P.; Supervision: D.J.K.; Validation: D.J.K.; Visualization: D.J.K.; Writing – original draft: D.J.K., A.G., A.P., A.R., W.D.; Writing – review & editing: D.J.K., A.G., A.P., A.R., W.D.

### Funding

This work was supported by the National Science Foundation (D.J.K.) and National Cancer Institute (NCI 1R01CA193473 to D.J.K.). Any opinion, findings, and conclusions or recommendations expressed in this material are those of the author(s) and do not necessarily reflect the views of the National Science Foundation or National Cancer Institute. Open Access funding provided by West Virginia University. Deposited in PMC for immediate release.

### Data and resource availability

All relevant data and details of resources can be found within the article and its supplementary information.

### Peer review history

The peer review history is available online at https://journals.biologists.com/bio/lookup/doi/10.1242/bio.062175.reviewer-comments.pdf

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
