## [Peer Review File · Biology Open]

A functional comparison of two transplantable syngeneic mouse models of melanoma: B16F0 and YUMM1.7

Alanna Gould, Anika Pirkey, Atefeh Razazan, Wentao Deng and David J. Klinke
DOI: 10.1242/bio.062175

Editor: Christopher Maher

Review timeline

Original submission:	21 July 2025
Editorial decision:	26 July 2025
First revision received:	7 August 2025
Accepted:	15 August 2025

Original submission

First decision letter

MS ID#: bio.062175

MS Title: A functional comparison of two transplantable syngeneic mouse models of melanoma: B16F0 and YUMM1.7

Authors: David J Klinke; Alanna Gould; Anika Pirkey; Atefeh Razazan; Wentao Deng
Article Type: Research Article

Dear Dr Klinke,

I have now reached a decision on the above manuscript.

The reviewer reports are shown at the bottom of this email or can be accessed, together with a copy of this decision letter, by going to:

As you will see, the reviewers gave favourable reports, but raised some critical points that will require amendments to your manuscript. I hope that you will be able to carry these out, because we would like to be able to accept your paper.

Reviewer 1

The study titled "A functional comparison of two transplantable syngeneic mouse models of melanoma: B16F0 and YUMM1.7" by Deng et al. offers a detailed and practical evaluation of two widely used murine melanoma models. Unlike earlier works that focused either on genetic characterization (Meeth et al., 2016) or on specific molecular regulators such as CCN4 (Deng et al., 2020), this paper centers exclusively on a side-by-side functional comparison of B16F0 and YUMM1.7 tumors under identical experimental conditions. The authors assess differences in tumor growth kinetics, angiogenic activity, immune cell infiltration, MHC-I expression, cytokine gene transcription, and sensitivity to antigen-specific CD8⁺ T cells using Pmel-1 transgenic TCR adoptive cell therapy. Their findings reveal that B16F0 tumors grow more rapidly, exhibit higher angiogenic factor expression, and are more sensitive to T cell-mediated killing, but also display extensive necrosis and ulceration. In contrast, YUMM1.7 tumors grow more slowly, show greater

immune infiltration and higher MHC-I levels, yet paradoxically resist T cell-mediated cytotoxicity. This contrast highlights distinct tumor-immune dynamics and demonstrates the unique strengths of each model: B16F0 may be more suitable for angiogenesis and drug response studies, while YUMM1.7 may be better suited for investigating tumor-immune interactions and resistance to immunotherapy. The paper's primary novelty lies in its integration of diverse phenotypic, immunological, and histological analyses into a unified comparative framework, offering actionable guidance for selecting appropriate preclinical models—an insight that is notably absent from prior comparative studies.

Despite these strengths, there are a few areas for improvement that should be addressed:

- * The manuscript would benefit from an expanded discussion on how findings from these murine models may translate to the heterogeneity of human melanoma, particularly regarding immune checkpoint resistance.
- * In Figure 5, it is unclear whether the replicates represent biological or technical repeats. This distinction is critical for assessing statistical robustness and reproducibility.
- * For Figures 4-6, consider the following improvements:
 - o Clearly report the number of biological replicates (n) in each figure legend.
 - o Show individual data points (e.g., dot plots) or use box-and-whisker plots to convey data distribution.
 - o Specify whether results reflect technical or biological replicates.
 - o Figures 4 and 6, in particular, appear to have tight or unreported error bars, suggesting low or unclarified replicate counts.
- * In the in vivo tumor growth methods section, the manuscript states that 6 mice per group were used. While acceptable, this is on the lower end for preclinical tumor studies, especially when measuring heterogeneous immune responses. If feasible, increasing the number of biological replicates would improve statistical power, or the authors should justify the sample size based on power calculations or prior literature.
- * The manuscript refers to the mouse strain as "C57BL/6," which is a generic label encompassing multiple sub-strains. The correct strain should be specified—either C57BL/6J (Jackson) or C57BL/6N (NIH)—due to known phenotypic and immunologic differences.
- * The abbreviation "NSG" should be expanded at first mention to ensure clarity: NOD scid gamma or NOD.Cg-Prkdcscid Il2rgtm1Wjl/SzJ.
- * The manuscript states that only female mice were used. Given well-documented sex differences in immune response and tumor biology, the authors should briefly justify this choice or acknowledge it as a limitation to broader applicability.
- * Given that the main strength of this study lies in its functional comparison of two murine melanoma models, I strongly suggest the addition of a summary table (either in the Results or Discussion section). This table could list and contrast the major findings related to tumor growth, necrosis, immune cell infiltration, angiogenesis, antigen presentation, and T cell sensitivity for B16F0 and YUMM1.7. Such a table would not only reinforce the practical utility of the manuscript but also improve accessibility and usability for readers selecting models for future experiments.

Overall, these are concerns that do not detract from the scientific merit of the work. The manuscript fills a significant gap in the melanoma modeling literature and will be a valuable resource for researchers designing preclinical studies in immuno-oncology.

Reviewer 2

statistical analysis: is missing from Figure 4 (far right column), recommend adding * to indicate level of significance and error bars.

reproducibility: needs to contain a materials and methods section that walks through the statistical analysis for Figure 1 (C-E).

Other Comments:

1. Expand more on the importance of BRAF mutation's presence in YUMM1.7 cell line in the introduction. This mutation is found in about half of melanoma cases and so having a cell line with this mutation gives researchers a more realistic model. I recommend just drawing the connection a little more clearly, since it is very important.

2. Results Section 1/Figure 1: "B16F0 tumors grew rapidly, reaching 1000 mm³ 96 in diameter within two weeks in C57BL/6 97 and three weeks in NSG." ----I believe these results need to be flipped. It looks like based on the figure that the it took 2 weeks for NSG (immunocompromised mice) and 3 weeks in B6 (immunocompetent mice). This would also be consistent with your description above this sentence. This figure should have a cut off line to indicate what data point you are referring to.

3. Results Section 1: "Both of these findings are consistent with literature citing that genetically engineered models are typically less immunogenic than spontaneous tumor models induced by carcinogens." ----Based on the introduction, it is not clear that the YUMM1.7 model is genetically engineered although that is what is being said in this section. Recommend elaborating on this more in the introduction if it is going to be revisited in results.

4. Results Section 1: Differences between YUMM1.7 cell line in NSG and B6 should be unpacked. You clearly walk through the results with B16F0 cell line, but it is lacking for YUMM1.7.

5. Results Section 2: "It is interesting to note that iCRT14 exhibited a similar level of toxicity in both cell lines as PLX-4720 had on cells harboring the activated BRAF mutation" ----Go ahead and just list the cell line that contains the mutation to keep reader on track. You can list YUMM1.7 which contains BRAF mutation.

Overall, I think this paper is really great and adds a really important piece to how we conduct research in animal models and the practicality of what we see in humans compared to animal studies. Your lab does a great job at emphasizing that importance and clearly showing it in your data and conclusions drawn. This paper has solid research experimental design and controls and the data was collected in a clear and concise manner that logically flows to the conclusions reached. It was a really fun paper to read and I have a few minor edits and revisions to make it a stronger paper.

Reviewer's Responses to Questions

Experimental quality

Does each figure have the proper controls?

If 'No', please indicate reasons in Comments for Author box below.

Reviewer #1:

- Yes

Reviewer #2:

- Yes

Were the data analyzed using appropriate statistical tests?

If 'No', please indicate reasons in Comments for Author box below.

Reviewer #1:

- Yes

Reviewer #2:

- No

Reproducibility

Were experiments performed using adequate number of biological replicates?

If 'No', please indicate reasons in Comments for Author box below.

Reviewer #1:

- Yes

Reviewer #2:

- Yes
-

Does the methods section provide sufficient detail to permit reproducibility?

If 'No', please indicate reasons in Comments for Author box below.

Reviewer #1:

- Yes

Reviewer #2:

- No
-

Completeness

Are the manuscript's conclusions supported by the data?

If 'No', please indicate reasons in Comments for Author box below.

Reviewer #1:

- Yes

Reviewer #2:

- Yes
-

Scholarship

Do the authors cite and discuss the merits of data that would argue for and against their conclusion?

If 'No', please indicate reasons in Comments for Author box below.

Reviewer #1:

- Yes

Reviewer #2:

- Yes
-

Does the manuscript title & abstract accurately reflect the contents of the manuscript, without hyperbole?

If 'No', please indicate reasons in Comments for Author box below.

Reviewer #1:

- Yes

Reviewer #2:

- Yes
-

First revision

Author response to reviewers' comments

A functional comparison of two transplantable syngeneic mouse models of melanoma: B16F0 and YUMM1.7

David J. Klinke II, Alanna Gould, Anika Pirkey, Atefeh Razazan, and Wentao Deng

Submitted for review in Biology Open (Manuscript ID: bio.062175)

Overall, we thank the reviewers for their service and their constructive criticisms on the manuscript. We feel that this revised paper presents, in a more comprehensive way, our functional comparison between these two mouse models of melanoma. Our responses to the specific comments are listed below in blue font. A red font is used to indicate the text that has been changed in this revised manuscript.

Reviewer 1:

General Comments:

The study titled "A functional comparison of two transplantable syngeneic mouse models of melanoma: B16F0 and YUMM1.7" by Deng et al. offers a detailed and practical evaluation of two widely used murine melanoma models. Unlike earlier works that focused either on genetic characterization (Meeth et al., 2016) or on specific molecular regulators such as CCN4 (Deng et al., 2020), this paper centers exclusively on a side-by-side functional comparison of B16F0 and YUMM1.7 tumors under identical experimental conditions. The authors assess differences in tumor growth kinetics, angiogenic activity, immune cell infiltration, MHC-I expression, cytokine gene transcription, and sensitivity to antigen-specific CD8⁺ T cells using Pmel-1 transgenic TCR adoptive cell therapy. Their findings reveal that B16F0 tumors grow more rapidly, exhibit higher angiogenic factor expression, and are more sensitive to T cell-mediated killing, but also display extensive necrosis and ulceration. In contrast, YUMM1.7 tumors grow more slowly, show greater immune infiltration and higher MHC-I levels, yet paradoxically resist T cell-mediated cytotoxicity. This contrast highlights distinct tumor-immune dynamics and demonstrates the unique strengths of each model: B16F0 may be more suitable for angiogenesis and drug response studies, while YUMM1.7 may be better suited for investigating tumor-immune interactions and resistance to immunotherapy. The paper's primary novelty lies in its integration of diverse phenotypic, immunological, and histological analyses into a unified comparative framework, offering actionable guidance for selecting appropriate preclinical models—an insight that is notably absent from prior comparative studies.

Despite these strengths, there are a few areas for improvement that should be addressed:

Overall, these are concerns that do not detract from the scientific merit of the work. The manuscript fills a significant gap in the melanoma modeling literature and will be a valuable resource for researchers designing preclinical studies in immuno-oncology.

Major Concerns

1. The manuscript would benefit from an expanded discussion on how findings from these murine models may translate to the heterogeneity of human melanoma, particularly regarding immune checkpoint resistance.
 Author Response (AR): We have expanded the discussion to include comments about how these results may translate to human melanoma. Clinical response to immune checkpoint therapies is predicated on the presence of cytolytic T lymphocytes within the tumor microenvironment. Here the reduced TILs in the B16F0 model suggests that it is not likely to respond to immune checkpoint therapies, as has been reported (PMID: 35481286, PMID: 25252955).
2. In Figure 5, it is unclear whether the replicates represent biological or technical repeats. This distinction is critical for assessing statistical robustness and reproducibility.
 AR: To clarify, TILs were isolated from tumors excised from tumor bearing mice. All mice bore a single tumor and 4 mice bearing B16F0 tumors and 3 mice bearing YUMM1.7 tumors were used. These are considered biological replicates. TIL aliquots were then stained with three different antibody panels that quantified T cells, B/NK cells, and myeloid cells in TIL isolates. All three antibody panels include the Live cell and CD45 markers so we have

reported the Live CD45+ fraction in each aliquot. These can be considered technical replicates. So in panel A, Live CD45+ values for both technical and biological replicates are shown (B16F0 n = 3 technical replicates of 4 biological replicates; YUMM1.7 n = 3 technical replicates of 3 biological replicates). The biological replicates have different x-values as the tumor sizes were not exactly the same among mice. To assess whether there is a difference in Live CD45+ values between B16F0 and YUMM1.7 tumors, we averaged out the technical replicates and just used averaged values that correspond to the biological replicates. The p-value was calculated using two-sided Student's t-test with equal variance. We have added some additional clarification to the caption.

3. For Figures 4-6, consider the following improvements:
 - Clearly report the number of biological replicates (n) in each figure legend.
 - Show individual data points (e.g., dot plots) or use box-and-whisker plots to convey data distribution.
 - Specify whether results reflect technical or biological replicates.
 - Figures 4 and 6, in particular, appear to have tight or unreported error bars, suggesting low or unclarified replicate counts.

AR: To clarify, the caption for Figure 4 states that the results shown are for three biological replicates. The distributions on the left show that the distributions are unimodal, which can be summarized by a central tendency statistic like a mean. The dot plots on the right show the results from all three biological replicates. ANOVA with post-hoc Tukey tests were used to assess statistical significance, where (a, b, c, or d) denote statistically different groups. It is not surprising that these biological replicates are tightly distributed as they are cell lines plated in different wells in a dish.

We have added some additional text to Figure 6 caption to clarify. The plot summarizes results obtained from each cell line secretome analyzed using a single antibody array. Each antibody probe is spotted in duplicate on the array (a technical replicate) and positive and negative controls are also on the array. The positive and negative controls can be used to establish a null distribution. The dotted lines in Figure 6 enclose 95% of the distribution of values that might be observed if there is no difference between the two groups, that is the null distribution. The vertical distance from the average of the two antibody probe spots over the vertical distance from the horizontal axis to the dotted line is proportional to a z-score. We have now added to Figure 6 two gray shaded lines that indicate z-scores equal to 3 and -3. A Z-score greater than 3 or less than -3 is considered significant and merits further consideration.

4. In the in vivo tumor growth methods section, the manuscript states that 6 mice per group were used. While acceptable, this is on the lower end for preclinical tumor studies, especially when measuring heterogeneous immune responses. If feasible, increasing the number of biological replicates would improve statistical power, or the authors should justify the sample size based on power calculations or prior literature.

AR: To clarify, the goal here was to estimate the intrinsic in vivo growth rate parameter of each cell line in NSG mice (so 2 parameter values in total) and then quantify the impact that selective immunologic pressure has on each cell tumor model in vivo (these are 2 additional parameter values). We have added some additional detail in the methods section about the number of data points used to estimate these values. Specifically, we are using 38 data points measured across 5 mice in the YUMM1.7/NSG cohort (one intrinsic growth rate parameter), 60 data points measured across 5 mice in the YUMM1.7/C57BL6 cohort (one immunologic pressure rate parameter), 39 data points measured across 8 mice in the B16F0/C57BL6 cohort, and 35 data points measured across 5 mice in the B16F0/NSG cohort. Across all cohorts more than 35 data points are used to estimate a single rate parameter value (or an average of 7.8 time points per mouse). Since tumors grow exponentially, plotting the log of tumor size versus time simplifies the problem to a linear regression, where the slope is the desired rate parameter. The intercepts correspond to the bolus of cells injected into each mouse that initiate a tumor and are determined separately for each mouse. Since injecting the cells into the mouse is done by hand using a syringe while you are immobilizing the mouse with your other hand, this can be a significant source of heterogeneity between animals. Moreover, instead of just presenting a point estimate of the parameter values, the Markov Chain Monte Carlo approach allows us to quantify the

uncertainty in estimating the value of the rate parameter, that is the posterior distribution. This general approach is described in more detail in a prior publication (Pirkey et al. Cell Mol Bioeng 2023). It seems that the concern expressed here implies asking a different question: a yes-no question related to whether a difference in survival between two groups exists. This is a different statistical question than what we are trying to show, which is what are the growth rates of these two cell lines in vivo and our confidence in those estimates. Since we are not testing statistical yes-no hypothesis, power calculations are not really relevant.

5. The manuscript refers to the mouse strain as "C57BL/6," which is a generic label encompassing multiple sub-strains. The correct strain should be specified—either C57BL/6J (Jackson) or C57BL/6N (NIH)—due to known phenotypic and immunologic differences..
AR: The particular mouse strain purchased from Charles River Labs is now specified in the text.
6. The abbreviation "NSG" should be expanded at first mention to ensure clarity: NOD scid gamma or NOD.Cg-Prkdcscid Il2rgtm1Wjl/SzJ..
AR: The specific NSG strain purchased from Jackson Lab is now specified in the text at first mention.
7. The manuscript states that only female mice were used. Given well-documented sex differences in immune response and tumor biology, the authors should briefly justify this choice or acknowledge it as a limitation to broader applicability.
AR: As we mention in the introduction, both the B16 and YUMM1.7 cell lines were generated in male mice. We have added some additional text to help clarify why we are using female mice. In theory, sex mismatch in transplanting male cells containing HY antigens in female C57BL/6 mice enhances the immunogenicity of the tumor. One of the common barriers cited for engaging anti-tumor immunity is that tumor cells lack antigens that the immune system recognize as foreign. By transplanting male cells in a female host, the system is design to have foreign antigens, such as HY antigen. If the immune system is unable to control for tumor growth, then there are other factors at work besides the lack of foreign antigen. Interestingly both B16F0 and YUMM1.7 are not controlled but it seems for different reasons.
8. Given that the main strength of this study lies in its functional comparison of two murine melanoma models, I strongly suggest the addition of a summary table (either in the Results or Discussion section). This table could list and contrast the major findings related to tumor growth, necrosis, immune cell infiltration, angiogenesis, antigen presentation, and T cell sensitivity for B16F0 and YUMM1.7. Such a table would not only reinforce the practical utility of the manuscript but also improve accessibility and usability for readers selecting models for future experiments..
AR: We have now included a summary table in the revised manuscript.

Reviewer 2:

General Comments:

Overall, I think this paper is really great and adds a really important piece to how we conduct research in animal models and the practicality of what we see in humans compared to animal studies. Your lab does a great job at emphasizing that importance and clearly showing it in your data and conclusions drawn. This paper has solid research experimental design and controls and the data was collected in a clear and concise manner that logically flows to the conclusions reached. It was a really fun paper to read and I have a few minor edits and revisions to make it a stronger paper.

Major Concerns

1. Reviewer 2: statistical analysis: is missing from Figure 4 (far right column), recommend adding * to indicate level of significance and error bars.
AR: To clarify, there is a statistical analysis presented in this far right column in Figure 4, where the letters "a, b, c and d" were used to indicate different groups. Of note, this is

not a pairwise comparison but an ANOVA with a post-hoc Tukey test was used to assess statistical significance between all four groups. Error bars are shown.

2. reproducibility: needs to contain a materials and methods section that walks through the statistical analysis for Figure 1 (C-E)..
AR: In the interest of brevity, we cited a previous paper (Pirkey et al. Cell Mol Bioeng 2023) that describes the statistical analysis of the tumor growth trajectories in more detail. In addition, it directs the reader to a GitHub repository that has the R code used in the analysis: <https://github.com/arcoolbaugh/B16-In-Vivo-Screen>

Other Concerns

1. Expand more on the importance of BRAF mutation's presence in YUMM1.7 cell line in the introduction. This mutation is found in about half of melanoma cases and so having a cell line with this mutation gives researchers a more realistic model. I recommend just drawing the connection a little more clearly, since it is very important.
AR: We have added text to draw this connection more clearly.
2. Results Section 1/Figure 1: "B16F0 tumors grew rapidly, reaching 1000 mm³ 96 in diameter within two weeks in C57BL/6 97 and three weeks in NSG." ----I believe these results need to be flipped. It looks like based on the figure that the it took 2 weeks for NSG (immunocompromised mice) and 3 weeks in B6 (immunocompetent mice). This would also be consistent with your description above this sentence. This figure should have a cut off line to indicate what data point you are referring to.
AR: Yes, thank you for the correction. The cutoff line is indicated by the gray shaded line at 1000 mm³.
3. Results Section 1: "Both of these findings are consistent with literature citing that genetically engineered models are typically less immunogenic than spontaneous tumor models induced by carcinogens." ----Based on the introduction, it is not clear that the YUMM1.7 model is genetically engineered although that is what is being said in this section. Recommend elaborating on this more in the introduction if it is going to be revisited in results.
AR: We have added more text to elaborate on the fact that the YUMM1.7 line is from a genetically engineered mouse model.
4. Results Section 1: Differences between YUMM1.7 cell line in NSG and B6 should be unpacked. You clearly walk through the results with B16F0 cell line, but it is lacking for YUMM1.7.
AR: We have added text to better walk through the results with the YUMM1.7 cell line.
5. Results Section 2: "It is interesting to note that iCRT14 exhibited a similar level of toxicity in both cell lines as PLX-4720 had on cells harboring the activated BRAF mutation" ----Go ahead and just list the cell line that contains the mutation to keep reader on track. You can list YUMM1.7 which contains BRAF mutation.
AR: We have added the clarifying text.

Second decision letter

MS ID#: bio.062175R1

MS Title: A functional comparison of two transplantable syngeneic mouse models of melanoma: B16F0 and YUMM1.7

Authors: David J Klinke; Alanna Gould; Anika Pirkey; Atefeh Razazan; Wentao Deng

Dear Dr Klinke,

I am happy to tell you that your manuscript has been accepted for publication in Biology Open, pending our standard publication integrity checks. It was accepted on 15 Aug 2025.